# Dihydroconiferyl Ferulate Isolated from *Dendropanax morbiferus* H.Lév. Suppresses Stemness of Breast Cancer Cells via Nuclear EGFR/c-Myc Signaling

**DOI:** 10.3390/ph15060664

**Published:** 2022-05-26

**Authors:** Yu-Chan Ko, Ren Liu, Hu-Nan Sun, Bong-Sik Yun, Hack Sun Choi, Dong-Sun Lee

**Affiliations:** 1Interdisciplinary Graduate Program in Advanced Convergence Technology and Science, Jeju National University, Jeju 63243, Korea; uchan@jejunu.ac.kr (Y.-C.K.); liuren0308@gmail.com (R.L.); 2Stem Cell Therapy and Regenerative Biology Laboratory, College of Life Science and Biotechnology, Heilongjiang Bayi Agricultural University, Daqing 163319, China; sunhunan76@163.com; 3Division of Biotechnology, College of Environmental and Bioresource Sciences, Jeonbuk National University, Gobong-ro 79, Iksan 54596, Korea; bsyun@jbnu.ac.kr; 4Subtropical/Tropical Organism Gene Bank, Jeju National University, Jeju 63243, Korea; 5Jeju Microbiome Research Center, Jeju National University, Jeju 63243, Korea; 6Faculty of Biotechnology, College of Applied Life Sciences, SARI, Jeju National University, Jeju 63243, Korea

**Keywords:** breast cancer stem cells, dihydroconiferyl ferulate, nuclear EGFR, Stat3, c-Myc

## Abstract

Breast cancer is the leading cause of global cancer incidence and breast cancer stem cells (BCSCs) have been identified as the target to overcome breast cancer in patients. In this study, we purified a BCSC inhibitor from *Dendropanax morbiferus* H.Lév. leaves through several open column and high-performance liquid chromatography via activity-based purification. The purified cancer stem cell (CSC) inhibitor was identified as dihydroconiferyl ferulate using nuclear magnetic resonance and mass spectrometry. Dihydroconiferyl ferulate inhibited the proliferation and mammosphere formation of breast cancer cells and reduced the population of CD44^high^/CD24^low^ cells. Dihydroconiferyl ferulate also induced apoptosis, inhibited the growth of mammospheres and reduced the level of total and nuclear EGFR protein. It suppressed the EGFR levels, the interaction of Stat3 with EGFR, and c-Myc protein levels. Our findings show that dihydroconiferyl ferulate reduced the level of nuclear epidermal growth factor receptor (EGFR) and induced apoptosis of BCSCs through nEGFR/Stat3-dependent c-Myc deregulation. Dihydroconiferyl ferulate exhibits potential as an anti-CSC agent through nEGFR/Stat3/c-Myc signaling.

## 1. Introduction

Breast cancer develops from breast tissue and is ranked fifth in mortality for cancers globally (6.9% of total cancer-related deaths) [1]. In women, breast cancer is the most commonly diagnosed and the leading cause of cancer-related deaths [1]. Breast cancer expresses well-characterized immunohistochemical markers (ER, PR, and HER2), proliferation markers (Ki-67), genomic markers (BRCA1, BRCA2, and PIK3CA), and immunologic markers (tumor-infiltrating lymphocytes) [2]. It is a very heterogeneous disease and may be divided into several major subcategories, including hormone-receptor (HR)-positive (HR+; ER+, PR+/−, and HER2−), HER2-positive (HER2+), and triple-negative (TN; ER−, PR−, and HER2−) breast cancer [3]. Compared with patients with other subtypes, TN breast cancer (TNBC) patients exhibit chemoresistance, recurrence, and metastasis [4]. Chemoresistance of breast cancer is associated with the presence of stem-like cells (cancer stem cells (CSCs)) [5]. Breast cancer stem cells (BCSCs) exhibit a capacity for self-renewal and are capable of initiating tumor growth. BCSC markers are designated as CD44^high^/CD24^low^ [6]. Several signaling pathways and transcription factors of breast CSCs are over-activated, compared with normal breast cancer cells. For example, the Wnt/β-catenin pathway is associated with cancer development and is believed to play an important role in the pathogenesis of hepatoblastoma, stem cell self-renewal, and CSC formation [7,8]. Notch signaling plays an important role in the maintenance of stemness of breast cancer cells [9]. The knockdown of tafazzin (TAZ, Hippo pathway) inhibits the self-renewal of BCSCs [10]. Targeting Stat pathways using sulforaphane inhibits endocrine-resistant, stem-like cells in breast cancer [11]. Myc inhibition depletes cancer stem-like cells in triple-negative breast cancer [12]. Interleukin 6 (IL-6) increases the invasive ability, metastatic potential, and stem-like phenotype of breast cancer cells [13]. Interleukin 8 (IL-8) is upregulated in breast cancer and is associated with a high risk of developing large, high-grade tumors [14].

The epidermal growth factor receptor (EGFR) is a membrane-bound receptor tyrosine kinase. It is a member of the ErbB family, which includes EGFR/ErbB-1/HER-1, ErbB-2/HER-2/neu, ErbB-3/HER-3, and ErbB-4/HER-4. ErbB-3 is the only member that does not exhibit tyrosine kinase activity [15]. EGFR is involved in multiple signaling pathways, including the PI3K/AKT, RAS/Raf/MAPK, JNK/STAT, and PLCγ/PKC pathways, which are associated with cell proliferation, DNA replication, and poor clinical outcomes [16,17,18,19]. Over the last few decades, many studies have demonstrated EGFR overexpression in TNBCs and its internalization from the cell membrane to the nucleus [20,21,22]. The translocated nuclear EGFR (nEGFR), which occurs through clathrin-coated pit-mediated receptor internalization, functions as a transcription factor to regulate the properties of CSCs through nuclear EGFR–PKM2 complexes [23,24]. These studies suggest that targeting nEGFR signaling is important for treating TNBC-derived stem cells.

*Dendropanax morbiferus* H.Lév. is a flowering plant from the family of *Araliaceae* and is known as a traditional medicinal plant in Korea, China and South America. The leaves, bark, roots, and stems of *Dendropanax morbiferus* H.Lév. are known as a traditional medicine for the prevention of several disorders [25]. The edible parts of *Dendropanax morbiferus* H.Lév. have been used as food additives [25]. The active compounds derived from the *Dendropanax* genus exhibit a broad range of therapeutic applications, including antioxidant, antibacterial, cytotoxic, anti-inflammatory, neuroprotective, antidiabetic, and anticancer properties [25,26,27,28,29]. Because *Dendropanax morbiferus* H.Lév. has pharmacological activities and is beneficial to human health, we isolated an inhibitor against breast CSCs from *Dendropanax morbiferus* H.Lév. on the basis of a mammosphere formation-guided assay. The isolated compound was identified as dihydroconiferyl ferulate, and it inhibits the formation of breast cancer mammospheres. In the present study, the isolated compound was identified as dihydroconiferyl ferulate, which suppresses mammosphere formation in breast cancer cell lines. This study also investigated whether dihydroconiferyl ferulate inhibited breast CSC formation through the EGFR/Stat3/c-Myc signaling pathway.

## 2. Results

### 2.1. Dihydroconiferyl Ferulate Isolated from Dendropanax morbiferus H.Lév. Suppresses the Mammosphere Formation Rate of Breast Cancer Cells

We purified a breast CSC inhibitor from *Dendropanax morbiferus* H.Lév. via fractionation guided with a mammosphere formation assay of MDA-MB-231 cells. Figure 1A shows the purification procedure. The extracted samples were purified using organic solvent extraction (methanol extraction and ethyl acetate extraction), silica gel, Sephadex LH-20 gel, preparatory TLC, and preparatory HPLC. The isolated compound suppressed the tumorsphere formation of MDA-MB-231 cells (Figure 1B). HPLC data indicated a high purity of the isolated compound (Figure 1C), which was identified as dihydroconiferyl ferulate using NMR and mass data (Figure 2 and Appendix A).

To determine whether dihydroconiferyl ferulate has a potent inhibitory effect on human breast cancer cells, we first tested the anti-proliferative effect on dihydroconiferyl ferulate at various concentrations in MCF-7 and MDA-MB-231 cells. Dihydroconiferyl ferulate inhibited the proliferation of MDA-MB-231 (Figure 3A) and MCF-7 (Figure 3B) cells at 75 µM. The IC50 value (the drug concentration required for 50% growth reduction in the survival curve) of MCF-7 and MDA-MB-231 cells was 112.4 μM and 114.6 μM, (Figure 3A,B) respectively. The results indicated that dihydroconiferyl ferulate suppresses breast cancer cell proliferation

To find the dihydroconiferyl ferulate inhibitor that possesses a CSC-suppressing effect, we carried out a mammosphere formation assay by automatic counting using the NICE program. To determine whether dihydroconiferyl ferulate can inhibit mammosphere formation, we treated the mammospheres with dihydroconiferyl ferulate. As shown in Figure 3C,D, dihydroconiferyl ferulate decreased the sphere size and number of tumorspheres derived from MDA-MB-231 and MCF-7 cells (Figure 3C,D). The results indicated that dihydroconferyl ferulate suppresses mammosphere formation.

Since cell colony formation and cell migration are likely to be two important processes in breast cancer tumorigenesis and metastasis, the colony formation and wound-healing assays were performed. As shown in Figure 3E,F, 50 μM of dihydroconiferyl ferulate inhibited colony formation and migration of MDA-MB-231 and MCF-7 cells. Curcumin and dihydroconiferyl ferulate have similar effects on migration and colony formation of breast cancer. The results indicate that dihydroconiferyl ferulate is a strong suppressor of breast cancer cell migration and colony formation.

### 2.2. Dihydroconiferyl Ferulate Suppresses CD44^high^/CD24^low^ Expressing Cells and Mammosphere Growth and Induces Mammosphere Apoptosis

The CD44^high^/CD24^low^ subpopulations are breast CSC populations and are a representative marker of breast CSCs. We cultured MDA-MB-231 cells in 6-well culture plates and then treated them with/without dihydroconiferyl ferulate for 24 h and determined the CD44^high^/CD24^low^ population in MDA-MB-231 breast cancer cells. Dihydroconiferyl ferulate decreased the CD44^high^/CD24^low^ population of breast cancer cells from 80.8% to 34.0% (Figure 4A).

To determine whether dihydroconiferyl ferulate induces apoptosis of breast CSCs and inhibits mammosphere growth, 5 day cultured mammospheres were treated with or without dihydroconiferyl ferulate (50 µM). The results indicated that dihydroconiferyl ferulate induced breast CSC apoptosis (Figure 4B).

To examine whether dihydroconiferyl ferulate regulates CSC-specific gene expression, we examined the transcription of CSC-specific genes using RT-qPCR. The dihydroconiferyl ferulate decreased c-Myc gene expression (Figure 4C). To confirm that dihydroconiferyl ferulate inhibits the proliferation of mammospheres, we incubated dihydroconiferyl ferulate on mammospheres and counted the cell numbers. Dihydroconiferyl ferulate reduced the cell number of dihydroconiferyl ferulate-incubated mammospheres and killed cells of mammospheres. These data indicated that dihydroconiferyl ferulate leads to a dramatic reduction in mammosphere growth (Figure 4D).

### 2.3. Dihydroconiferyl Ferulate Reduces the Total and Nuclear Levels of EGFR in Breast CSCs

To identify the underlying molecular mechanism associated with the inhibition of mammosphere formation by dihydroconiferyl ferulate, we measured the total and nuclear levels of EGFR via Western blot analysis. The total and nuclear levels of EGFR protein were significantly reduced after a 48 h treatment with dihydroconiferyl ferulate (50 µM) (Figure 5A,B).

### 2.4. Dihydroconiferyl Ferulate Inhibits the Interaction of EGFR with Stat3 and Decreases the Total and Nuclear Levels of pStat3 and Stat3 in Breast CSCs

Breast cancer cells overexpress the EGFR protein and EGFR exhibits membrane-bound and nuclear signaling activities. Nuclear EGFR induces resistance to anti-EGFR therapy and is a therapeutic target for breast cancer. Nuclear EGFR (nEGFR) is also a regulator and cofactor of Stat3 [30,31,32,33]. We evaluated the interaction between Stat3 and EGFR protein using immunoprecipitation. Dihydroconiferyl ferulate inhibited the interaction of EGFR and Stat3 (Figure 6A). Subsequently, the total and nuclear levels of p-Stat3 and Stat3 protein were assessed using Western blot analysis. Dihydroconiferyl ferulate decreased the total level of p-Stat3 (Figure 6B), as well as cytosolic and nuclear p-Stat3 and Stat3 levels (Figure 6C).

### 2.5. Dihydroconiferyl Ferulate Decreases the Transcript Level of c-Myc and Inhibits the Total and Nuclear Levels of c-Myc Protein

Nuclear EGFR is known to function as a co-transcription factor of Stat3 to enhance the transcription of the c-myc gene [30]. We examined the inhibitory effect on c-Myc transcription through the EGFR-Stat3 complex, which was inhibited by dihydroconiferyl ferulate treatment. The relative mRNA and protein expression of c-Myc were determined following dihydroconiferyl ferulate (50 µM) exposure. As shown in Figure 7A,B, dihydroconiferyl ferulate decreased c-Myc mRNA levels, as well as the total and nuclear levels of c-Myc protein (Figure 7C,D). The downregulation of Stat3 also decreased the protein and mRNA expression levels of c-Myc (Figure 7E,F). Additionally, c-Myc knockdown inhibited mammosphere formation (Figure 7G). This suggests that dihydroconiferyl ferulate inhibits c-Myc expression via Stat3 and/or EGFR-Stat3 complex inhibition. Figure 8 shows a schematic illustration of the proposed mechanism through which dihydroconiferyl ferulate inhibits breast CSC formation via the EGFR-Stat3/c-Myc signaling pathway.

## 3. Discussion

*Dendropanax morbiferus* H.Lév. is a traditional medicinal plant that belongs to Araliaceae. Previous studies have shown that *Dendropanax morbiferus* H.Lév. exerts various biological activities, including antifungal [34], antibacterial [34], cytotoxic [35,36], antidiabetic [37], antioxidant [38,39], anti-inflammatory [40,41], hepatoprotective [42], neuroprotective [43,44], and anticancer [45] activities. The bioactive compounds contained in *Dendropanax morbiferus* H.Lév. have been summarized by R.B. et al. [25] and include flavonoids, pyrimidines, terpenoids, polyphenols, tannins, essential oils, alkaloids, and phenol carboxylic acids, such as lutexin, β-sitosterol, chlorogenic acid, protocatechuic acid, quercetin, trans-ferulic acid, caffeic acid, and resveratrol. The extract of Dendropanax morbifera Léveille caused an increase in apoptotic or senescent cells in Huh-7 cells and caused the activation of p16 and p53 pathways. The inhibition of Akt or ERK signaling resulted in the suppression of Huh-7 cell proliferation [26].

In the present study, we isolated a compound from *Dendropanax morbiferus* H.Lév. via multiple chromatography steps using activity-guided fractionation (Figure 1 and Appendix A). The resulting compound was identified as dihydroconiferyl ferulate via ESI-MS and NMR (Figure 2 and Appendix A). Subsequently, the inhibitory effect of dihydroconiferyl ferulate on breast cancer cell proliferation and mammosphere formation was demonstrated. As shown in Figure 3, dihydroconiferyl ferulate inhibits the proliferation of TNBC, MDA-MB-231, and ER+ MCF-7 cells, as well as mammosphere formation. The expression of the CD44^high^/CD24^low^ CSC phenotype and the effect of dihydroconiferyl ferulate on stemness in MDA-MB-231 cells was assessed. The results indicated that dihydroconiferyl ferulate decreases the CD44^high^/CD24^low^ subpopulation from 80.8% to 34% (Figure 4A). The apoptosis and growth of breast CSCs were determined following dihydroconiferyl ferulate (50 µM) treatment. As shown in Figure 4B,D, dihydroconiferyl ferulate inhibited breast cancer stem cell growth and induced apoptosis. The structure of dihydroconiferyl ferulate is similar to that of butein and curcumin. Butein is an active flavonoid isolated from the bark of *Rhus verniciflua* and exhibits anticancer activity against various human cancer cell types, including osteosarcoma, colon carcinoma, breast carcinoma, hepatocarcinoma, and lymphoma [46,47,48,49,50]. Previous studies have shown that butein induces an apoptotic effect in hepatic cells by decreasing Stat3-related gene expression [51]. Butein also suppresses the protein tyrosine kinase activity of EGFR in HepG2 cells [52] and circumvents gefitinib-resistant lung cancer growth by inhibiting EGFR [53]. Curcumin is a polyphenol compound that is isolated from *Curcuma longa* and exhibits therapeutic benefits in various types of cancers. The primary molecular targets of curcumin are Stat3, Stat5, Notch-1, NF-κB, VEGF, and interleukins [54]. Curcumin potentiates the anticancer activity of gefitinib both in vitro and in vivo through the inhibition of EGFR phosphorylation and induces EGFR ubiquitination in non-small cell lung cancer [55]. EGFR is also a co-transcription factor that regulates c-fos, cyclin D1, inducible nitric oxide synthase (iNOS), cyclooxygenase-2 (COX-2), and aurora kinase A [31,56,57,58]. EGFR regulates stemness in several cancer types and treatment of EGFR inhibitors regulates the appearance of stemness in lung cancer cells [59].

Using these data, we determined the level of total and nuclear EGFR protein in mammospheres following dihydroconiferyl ferulate treatment. The results indicated that dihydroconiferyl ferulate decreases the expression of both total and nuclear EGFR (Figure 5). Nuclear EGFR interacts with Stat3 to form a complex, which can regulate the expression of IL-6 and c-Myc [32]. Dihydroconiferyl ferulate inhibited the nEGFR/Stat3 complex (Figure 6A) and decreased the phosphorylation of Stat3 in total lysates prepared from mammospheres (Figure 6B). Moreover, dihydroconiferyl ferulate decreased the nuclear levels of pStat3 and Stat3 protein (Figure 6C). The inhibition of the nEGFR/Stat3 complex resulted in the downregulation of c-Myc transcription following dihydroconiferyl ferulate treatment (Figure 7A,B) and reduced the total and nuclear levels of c-Myc (Figure 7C,D). The knockdown of Stat3 using si-RNA inhibited the transcription and expression of c-Myc (Figure 7E,F), whereas the knockdown of c-Myc resulted in the inhibition of mammosphere formation (Figure 7G). *Dendropanax morbifera* extracts have anti-cancer effects on MCF-7 and MDA-MB-231 cells. However, our isolated and analyzed compound has anti-CSC effects. We also confirmed the mechanism of how this compound inhibits breast cancer stem cells. Our results showed that dihydroconiferyl ferulate has potential as an anti-CSC agent. Dihydroconiferyl ferulate inhibited the growth of breast cancer cells and mammospheres. Cell colony formation and cell migration are likely to be two important processes in breast cancer tumorigenesis and metastasis. Metastasis is known to cause high rates of recurrence and mortality in cancer patients. Our inhibitor suppressed the migration and colony formation of the breast cancer cells (Figure 3). In the present study, the isolated compound, dihydroconiferyl ferulate, suppresses mammosphere formation in breast cancer cell lines. Our data showed that dihydroconiferyl ferulate inhibited breast CSC formation through the EGFR/Stat3/c-Myc signaling pathway. Our data suggest that dihydroconiferyl ferulate provides a new compound for breast cancer therapy by targeting EGFR signaling.

## 4. Materials and Methods

### 4.1. Chemicals and Reagents

Silica gel 60 powder and silica gel 60 F_254_ aluminum sheets and glass plates for thin-layer chromatography (TLC) were obtained from Merck Supelco (Darmstadt, Hesse, Germany). Sephadex LH-20 (LH20_100) power was obtained from Millipore (Cytiva, Marlborough, MA, USA). High-pressure liquid chromatography (HPLC) was conducted using a Shimadzu LC-10 system (Tokyo, Japan). Breast cancer cell viability was determined using a cell viability assay kit (EZ-Cytox, DoGenBio, Seoul, Korea). The other chemicals and organic solvents were obtained from Sigma (St. Louis, MO, USA).

### 4.2. Plant Material Source

Dried *Dendropanax morbiferus* H.Lév. leaves were obtained from urban farmers (Seogwipo, Jeju, Korea). The *Dendropanax morbiferus* H.Lév. leaves were ground. A sample (no. 2020_011) was deposited at the Faculty of Biotechnology, Jeju National University (Jeju-Si, Korea). The authenticator of the plant material, such as the leaves of the Dendropanax morbiferus plant identified by Professor Yong-Suk Chung (Jeju National Univesity).

### 4.3. Isolation of an Inhibitor of Mammosphere Formation from Dendropanax morbiferus H.Lév.

For organic solvent extraction, the method for the isolation of a mammosphere formation inhibitor from *Dendropanax morbiferus* H.Lév. was described previously [60]. *Dendropanax morbiferus* H.Lév. powder (1000 g) was added to methanol and the mixture was extracted (MeOH 30 L). Figure 1A summarizes the isolation procedure. *Dendropanax morbiferus* H.Lév. (40 g) powder was extracted with 1.2 L of methanol in a 3 L flask (total 25 flasks) at 30 °C overnight, using a shaking incubator. The methanol extracts were filtered with filter paper (ADVANTEC^®^, Niigata, Japan) and evaporated from 30 L to 5 L for 10 h using a rotary evaporator (Heidolph, Schwabach, Germany). It was mixed with a 2-fold volume of water (*v/v* = 1:2), and the methanol was evaporated from the mixture at 55 °C. The water-suspended mixture was extracted with an equal volume of ethyl acetate (EA, *v*/*v* = 1:1) using a separatory funnel (Sigma-Aldrich, Burlington, MA, USA). The ethyl acetate was evaporated at 55 °C using a rotary evaporator for 30 min and then the extracts were dissolved with 200 mL of methanol.

For chromatography, the 10 mL of EA extract was evaporated at 55 °C and then concentrated with 5 mL of chloroform-methanol solvent (CHCl_3_: MeOH, 10:1). It was separated on a silica gel column (3 × 35 cm; particle size: 40–63 µm), and eluted with a chloroform-methanol solvent (CHCl_3_: MeOH, 10:1, Appendix A). We separated the samples about 20 times. The column was fractionated into six parts on the basis of color. The fractionated samples were evaporated at 55 °C and dissolved with methanol. Then, each fraction was subjected to a mammosphere formation assay. Fraction #4 strongly inhibited mammosphere formation and was further separated into four parts using a Sephadex LH-20 gel column (2.5 × 30 cm, particle size: 25–100 µm) with methanol (Appendix A). These four fractions were isolated and evaluated using the mammosphere formation assay. Fraction #3 exhibited significant inhibitory activity in the assay; thus, it was loaded onto a preparatory TLC silica gel glass plate (20 × 20 cm, Merck KGaA, Darmstadt, Germany) and developed in a glass chamber (CHCl_3_: MeOH, 10:1). The major band was scraped out with knife, dissolved with methanol, centrifuged for 5 min, and then concentrated with methanol. Its activity was examined using the mammosphere formation assay (Appendix A).

For HPLC, the active band was analyzed on a HPLC instrument (Shimadzu LC-20A, Kyoto, Japan) with an ODS column (10 × 250 mm, flow rate: 2 mL/min, mobile phase: acetonitrile-water). The acetonitrile concentration was initiated at 0%, increased to 60% at 10 min, reached 100% at 30 min, and eluted for 10 min (Appendix A). The major peak was collected and subjected to structural analysis.

### 4.4. Structure Analysis of the Isolated Compound

The chemical structure of the isolated compound was analyzed via mass spectrometry and NMR. The molecular weight was determined to be 358 by ESI-mass measurement, which showed quasi-molecular ion peaks at *m*/*z* 359.4 [M + H]^+^ and 381.3 [M + Na]^+^ in positive mode and at *m*/*z* 357.3 [M − H]^−^ in negative mode (see Appendix A). The ^1^H NMR spectrum measured in CD_3_OD exhibited signals because of 6 aromatic methine protons at δ 7.18 (d, *J* = 1.8 Hz), 7.06 (dd, *J* = 8.4, 1.8 Hz), 6.80 (d, *J* = 8.4 Hz), 6.77 (d, *J* = 1.8 Hz), 6.70 (d, *J* = 8.4 Hz), and 6.63 (dd, *J* = 8.4, 1.8 Hz), which are attributable to 2 1,2,4-trisubstituted benzenes, 2 olefinic methines at δ 7.57 (d, *J* = 16.2 Hz) and 6.35 (d, *J* = 16.2 Hz), which were assigned to a *trans*-1,2-disubstituted double bond, 1 oxygenated methylene at δ 4.16, 2 methylenes at δ 2.65 and 1.97, and 2 methoxy groups at δ 3.89 and 3.82 (see Appendix A). The ^13^C NMR spectrum in combination with the HMQC spectrum revealed 20 carbon peaks, including an ester carbonyl carbon at δ 169.4; 4 oxygenated sp^2^ carbons at δ 150.8, 149.4, 148.9, and 145.7; 8 sp^2^ methine carbons at δ 146.7, 124.1, 121.9, 116.5, 116.2, 115.5, 113.2, and 111.7; 2 sp^2^ carbons at δ 134.2 and 127.6; 3 methylene carbons at δ 64.9, 32.8, and 31.8; and 2 methoxy carbons at δ 56.4 and 56.3 (see Appendix A). The ^1^H-^1^H COSY spectrum established four partial structures, and two 1,2,4-trisubstituted benzenes, –CH_2_–CH_2_–CH_2_–, and –CH=CH– (see Appendix A). Further structural elucidation was performed using the HMBC spectrum, which showed a long-range correlation from the methylene protons at δ 2.65 to the carbons at δ 134.2, 121.9, and 113.2, from the methine proton at δ 7.57 to the carbons at δ 127.6, 124.1, and 111.7, and from the methine protons at δ 7.57 and 6.35 and the methylene protons at δ 4.16 to the carbonyl carbon at δ 169.4. Finally, 2 methoxy protons at δ 3.89 and 3.82 exhibited long-range correlations with the oxygenated sp^2^ carbons at δ 149.4 and 148.9, respectively (see Appendix A). Thus, the structure of the isolated compound was identified as dihydroconiferyl ferulate (Figure 2).

### 4.5. Culture of Human Breast Cancer Cells and Mammospheres

MDA-MA-231 and MCF-7 were purchased from the Korea Cell Line Bank (Seoul, Korea) and cultured in Dulbecco’s Modified Eagle’s Medium containing 10% fetal bovine serum (Corning, Glendalec, AZ, USA) and 1% penicillin/streptomycin solution (Corning) in an incubator with a 5% CO_2_ atmosphere. MDA-MA-231 (1 × 10^4^ per well) and MCF-7 (4 × 10^4^ per well) were incubated in a 24- or 6-well ultralow attachment plate with MammoCult^TM^ culture medium (StemCell Technologies, Vancouver, BC, Canada) containing hydrocortisone (0.5 μg/mL) and heparin (4 μg/mL) for 7 days.

### 4.6. Cell Proliferation and Mammosphere Formation Assay

MCF-7 (1.5 × 10^6^ cells/plate) and MDA-MB-231 (1 × 10^6^ cells/plate) cancer cells were seeded into 96-well plates for 24 h and incubated with dihydroconiferyl ferulate (0, 10, 25, 50, 75, 100, 150, and 200 µM) for 24 h. Subsequently, cell viability was examined using the EZ-Cytox Plus Kit (DoGenBio, Seoul, Korea), according to the manufacturer’s protocol. A VERSA_max_ ELISA reader (Molecular Device, San Jose, CA, USA) was used to measure the OD_450_. MDA-MA-231 (1 × 10^4^ per well) and MCF-7 (4 × 10^4^ per well) were incubated in a 6-well ultralow attachment plate with MammoCult^TM^ culture medium for 7 days. Mammospheres were scanned and analyzed using the NIST integrated colony enumerator program [61]. The rate of mammosphere formation was measured by determining the mammosphere formation efficiency (MFE) [62].

### 4.7. CD44^+^/CD24^−^ Expression and Apoptosis via Flow Cytometry and Mammosphere Growth Assay

MDA-MB-231 cancer cells (1.5 × 10^6^ cells) were cultured in a 6-well plate for 24 h and treated with DMSO as control or dihydroconiferyl ferulate (50 μM) for 24 h [63]. The cells were harvested, dissociated, and incubated with monoclonal antibody antihuman CD44 (FITC-conjugated) and monoclonal antibody antihuman CD24 (APC-conjugated) antibodies (BD) for 45 min at 4 °C. After washing with 1X PBS, the CD44^+^/CD24^−^ cell populations were examined using an Accuri C6 machine (BD San Jose, CA, USA). The Annexin V Apoptosis Detection kit with PI (BD) was used to measure apoptosis of the mammospheres treated with dihydroconiferyl ferulate (50 μM) following the manufacturer’s protocol. Mammospheres were collected and dissociated with 0.05% trypsin-EDTA 1X (Corning). Briefly, 1 × 10^6^ cells were incubated with Annexin V (FITC) and PI in a binding buffer at room temperature (RT), protected from light for 30 min, and the cells were examined via flow cytometry at the Jeju Center of Korea Basic Science Institute (core facility center, Jeju, South Korea). Mammospheres derived from MDA-MB-231 cells cultured with or without dihydroconiferyl ferulate (50 µM) were divided into single cells, and an equal number of cancer cells were seeded into six-well plates. The number of cells was counted at 24, 48, and 72 h to measure the growth of the mammospheres.

### 4.8. Western Blot Analysis

Western blot analysis was conducted on the basis of a previously described method [64]. Protein extracts of mammospheres derived from MDA-MB-231 cells (1 × 10^4^ per well) treated with dihydroconiferyl ferulate (50 μM) were analyzed using SDS-PAGE (8%) and electrotransferred to polyvinylidene difluoride membranes (Millipore, Billerica, MA, USA). The membranes were incubated at RT for 1 h in Odyssey blocking buffer (LI-COR, Lincoln, NB, USA) containing Tween 20 (0.1%, *v*/*v*). The membranes were then incubated overnight at 4 °C in a blocking buffer containing the following primary antibodies: anti-EGFR (#4267s, Cell Signaling Technology, Denver, CO, USA), anti-pStat3 (#9145s, Cell Signaling Technology, Denver, CO, USA), anti-c-Myc (#5605s, Cell Signaling Technology, Denver, CO, USA), anti-Stat3 (sc-482), anti-Lamin B (sc-6216), and anti-β-actin (sc-47778 Santa Cruz Biotechnology, Dallas, TX, USA). The membranes were washed with PBST (phosphate-buffered saline with Tween 20, 0.1%, *v*/*v*) and incubated with anti-rabbit (IRDye 800CW-conjugated), anti-goat (IRDye 680RD-conjugated), or anti-mouse (IRDye 680RD-conjugated) secondary antibodies. An ODYSSEY CLx machine (LI-COR, Lincoln, NB, USA) was used to examine the band signals.

### 4.9. Immunoprecipitation (IP)

Proteins extracts of mammospheres treated with DMSO as a control or dihydroconiferyl ferulate (50 μM) were prepared in lysis buffer (20 mM Hepes, 10 mM EGTA, 40 mM glycerol 2-phosphate, 2.5 mM MgCl_2_ 6H_2_O, 1% NP-40, pH 7.5). IP was performed using 1 μg of Stat3 antibody (sc-482) and 500 μg of protein. Protein A/G-Agarose (P9203-100, GenDEPOT) was used to precipitate the protein complex, which was then analyzed using SDS-PAGE, followed by immunoblotting with the EGFR antibody (#4267s).

### 4.10. RNA Extraction and qRT-PCR

RNA extraction and qRT-PCR were performed as described previously [64]. Total RNA was isolated from MDA-MB-231 cancer cells or mammospheres using the TaKaRa MiniBEST Universal RNA Extraction Kit according to the manufacturer’s protocol. We used the TOPreal^TM^ One-step RT-qPCR kit (SYBR Green with low ROX, Enzynomics, Daejeon, Korea) for performing qRT-PCR. All gene-specific primers were purchased from Bioneer (Daejeon, Korea). The β-actin gene was used as an internal control (Appendix A).

### 4.11. Stat3 and c-Myc Knockdown Using Small Interfering RNA (siRNA)

MDA-MB-231 cells (1 × 10^6^) were seeded into 6-well plates for 24 h and transfected with human c-Myc-specific siRNAs from Bioneer (Daejeon, Korea) for 48 h to confirm the effect of the c-Myc protein on mammosphere formation and with human Stat3-specific siRNA from Bioneer (Daejeon, Korea) to determine the effect of Stat3 on the expression of the c-Myc protein. We used Lipofectamine 2000 (Invitrogen, Carlsbad, CA, USA) for siRNA transfection of the breast cancer cells. Immunoblotting was performed to examine the expression levels of c-Myc and Stat3 following specific siRNA transfection.

### 4.12. Statistical Analysis

All data are shown as the mean ± standard deviation. The differences among the means were analyzed using a one-way ANOVA and a Student’s *t*-test; *p* values of <0.05 were considered statistically significant (GraphPad Prism 7 software).

## 5. Conclusions

Dihydroconiferyl ferulate isolated from *Dendropanax morbiferus* H.Lév. inhibits the proliferation of breast cancer cell lines and mammosphere formation. It reduces the population of CD44^high^/CD24^−^ cells, induces apoptosis, suppresses the proliferation of mammospheres, decreases total and nuclear EGFR, inhibits EGFR/Stat3 complex formation, nuclear Stat3 and pStat3 protein, and subsequently inhibits the transcription and expression of c-Myc. Our results show that dihydroconiferyl ferulate exhibits anti-BCSC activity through the nEGFR/Stat3/c-Myc signaling pathway.

## Figures and Tables

**Figure 1 pharmaceuticals-15-00664-f001:**
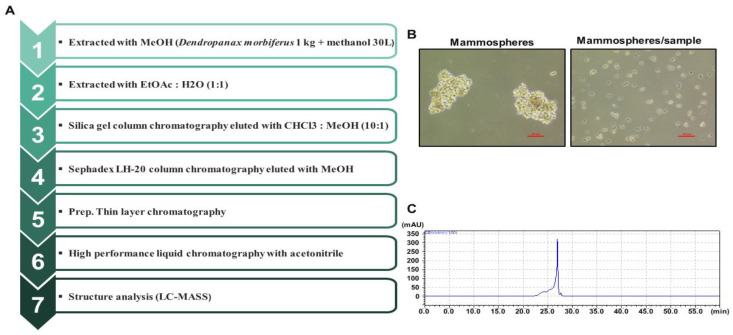
Purification of a breast CSC inhibitor from *Dendropanax morbiferus* H.Lév. using fractionation guided by a mammosphere assay. (**A**) Isolation protocol for the inhibitor of mammosphere formation from *Dendropanax morbiferus* H.Lév. (**B**) Effect of the purified inhibitor from *Dendropanax morbiferus* H.Lév. on mammosphere formation using MDA-MB-231 cells. Images were analyzed using microscopy at 10× magnification (scale bar = 100 µm). (**C**) HPLC data for the isolated samples from *Dendropanax morbiferus* H.Lév.

**Figure 2 pharmaceuticals-15-00664-f002:**
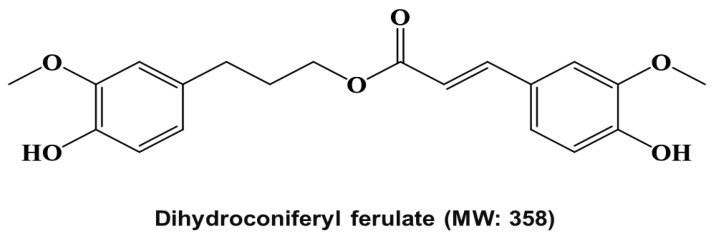
Molecular structure of the cancer stem cell inhibitor from *Dendropanax morbiferus* H.Lév.

**Figure 3 pharmaceuticals-15-00664-f003:**
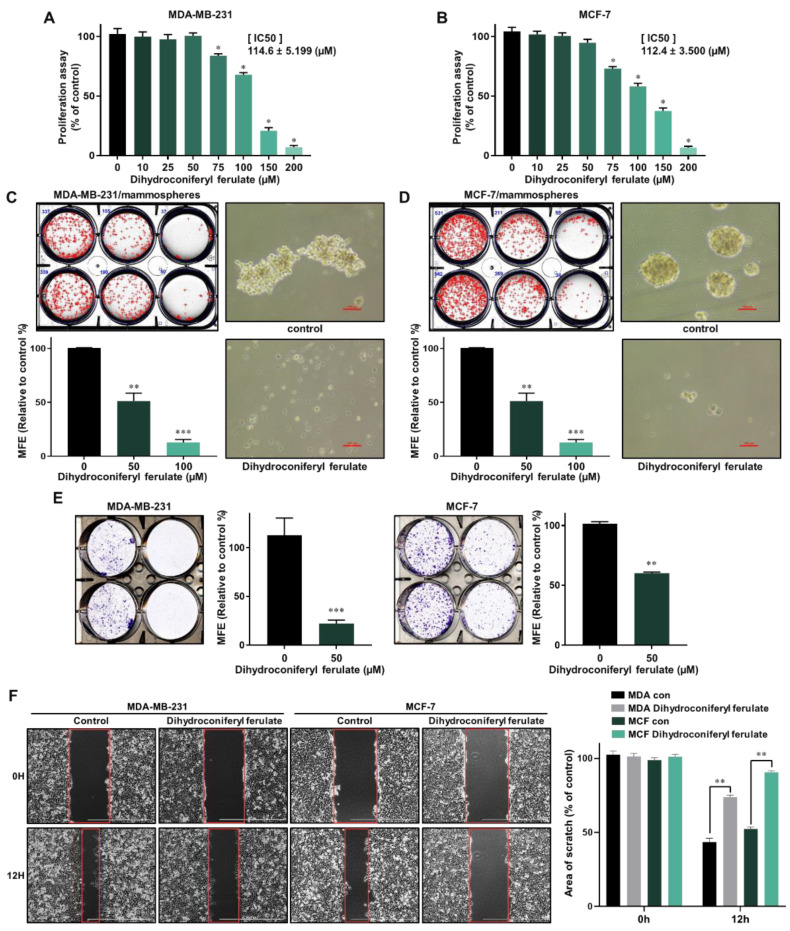
Inhibitory effects of dihydroconiferyl ferulate on cancer cell proliferation and formation of mammospheres. (**A**,**B**) The antiproliferative ability of dihydroconiferyl ferulate was determined using a proliferation assay kit with MDA-MB-231 and MCF-7 cells treated with dihydroconiferyl ferulate. * IC50 value is the drug concentration required for 50% growth reduction in the survival curve. (**C**,**D**) To measure the effect of dihydroconiferyl ferulate on the formation of mammospheres derived from breast cancer cells, they were cultured with increasing concentrations of dihydroconiferyl ferulate. After 7 days, the mammosphere morphologies were analyzed. Representative tumorsphere images were examined via inverted microscopy at 10× magnification (scale bar = 100 µm). (**E**,**F**) Effects of dihydroconiferyl ferulate on colony formation and migration of breast cancer cells. MDA-MB-231 and MCF-7 cells (1 × 10^3^ per well) were cultured with or without dihydroconiferyl ferulate for 7 days. The colonies were scanned using a scanner. Migration images for dihydroconiferyl ferulate were captured at 12 h (scale bar: 1000 µm). The percent inhibition of cancer cell migration was determined relative to the untreated control, which was designated 100. Representative data were collected and represent the mean ± standard deviation; ** *p* < 0.01, *** *p* < 0.001 vs. control.

**Figure 4 pharmaceuticals-15-00664-f004:**
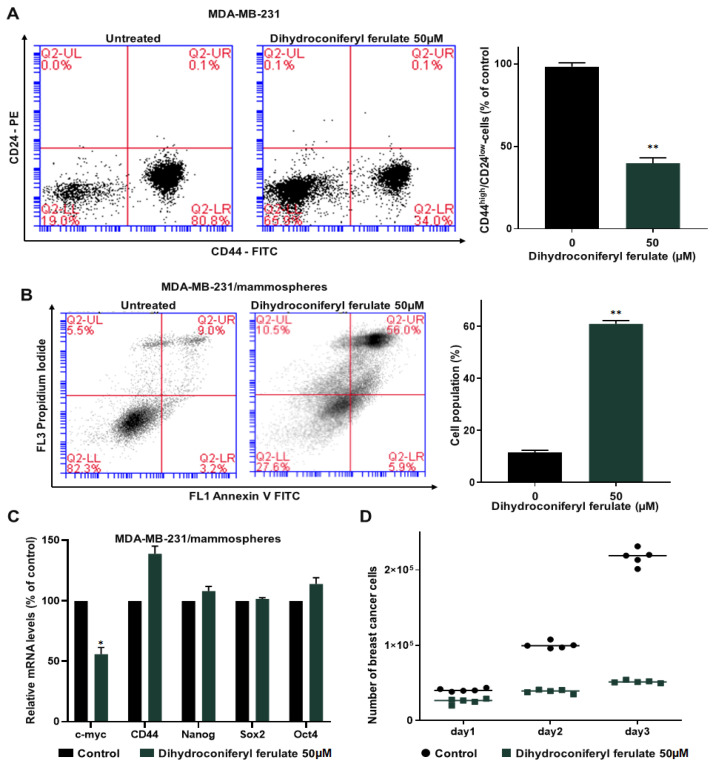
Dihydroconiferyl ferulate suppresses the subpopulations of CD44^high^/CD24^low^ and the growth of mammospheres and induces apoptosis of mammospheres. (**A**) MDA-MB-231 cells were treated with dihydroconiferyl ferulate (50 µM) for 24 h. The CD44^high^/CD24^low^ populations of breast cancer cells were examined by flow cytometry. (**B**) MDA-MB-231 tumorspheres were cultured for 5 days and then treated with dihydroconiferyl ferulate (50 µM) for 48 h. Tumorspheres were segregated into single cells and apoptosis was measured with an Annexin V/PI assay kit. (**C**) Transcripts for the CSC markers, Oct4, CD44, Sox2, c-myc, and Nanog, were examined in mammospheres treated with dihydroconiferyl ferulate using gene-specific primers by qRT-PCR (Appendix A). β-actin was used as a control. (**D**) Dihydroconiferyl ferulate suppresses mammosphere growth. Mammospheres treated with dihydroconiferyl ferulate were segregated into single cells and an equal number of cells were seeded into 6 cm culture dishes. The cells were counted at 24, 48, and 72 h. Representative data were collected and represent the mean ± standard deviation; * *p* < 0.05, ** *p* < 0.01 vs. control.

**Figure 5 pharmaceuticals-15-00664-f005:**
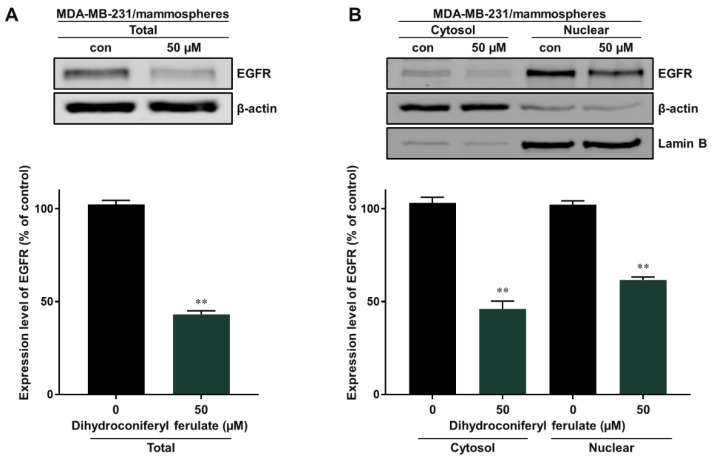
Inhibitory effect of dihydroconiferyl ferulate on EGFR signaling. (**A**) Western blot analysis showing the level of total EGFR measured in tumorspheres derived from MDA-MB-231 cells after treatment with dihydroconiferyl ferulate (50 µM) or DMSO for 48 h. (**B**) The nuclear protein levels of EGFR were examined in MDA-MB-231-derived mammospheres treated with dihydroconiferyl ferulate (50 µM). Dihydroconiferyl ferulate decreases the total and nuclear protein levels of EGFR. Data from the experiments represent the mean ± standard deviation; ** *p* < 0.01 vs. control.

**Figure 6 pharmaceuticals-15-00664-f006:**
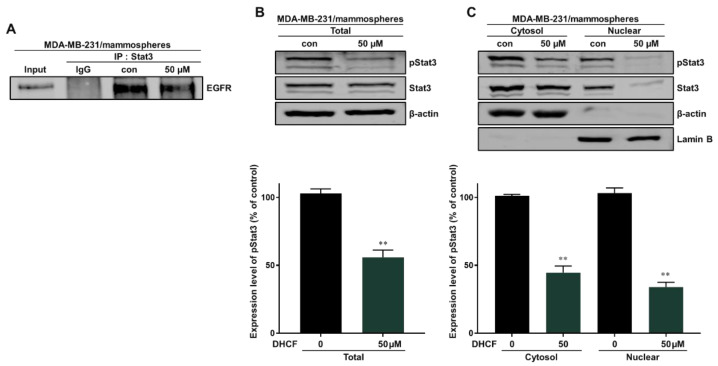
Dihydroconiferyl ferulate inhibits the interaction between EGFR and Stat3. (**A**) Stat3 and EGFR were immunoprecipitated with anti-Stat3 antibody from mammosphere extracts after treatment with dihydroconiferyl ferulate (50 µM) for 48 h. Immunoblotting was carried out using the anti-EGFR antibody. (**B**) Results of immunoblot analysis showing the levels of pStat3 and Stat3 in total proteins measured in mammosphere extracts after treatment with dihydroconiferyl ferulate (50 µM) for 48 h. (**C**) The nuclear levels of pStat3 and Stat3 proteins were determined in mammospheres derived from MDA-MB-231 treated with dihydroconiferyl ferulate (50 µM) or DMSO for 48 h. Dihydroconiferyl ferulate decreased the levels of p-Stat3 and Stat3 in mammospheres. Data from experiments represent the mean ± standard deviation; ** *p* < 0.01 vs. control.

**Figure 7 pharmaceuticals-15-00664-f007:**
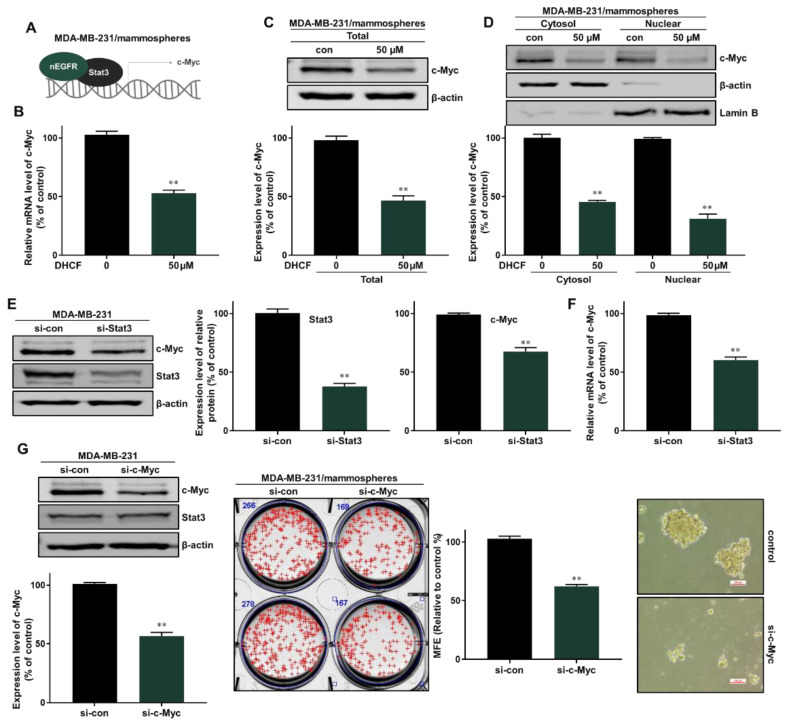
Dihydroconiferyl ferulate inhibits the c-Myc pathway via Stat3/nEGFR. (**A**,**B**) The relative mRNA levels of c-Myc were measured in mammospheres after treatment with dihydroconiferyl ferulate (50 µM) for 48 h. Dihydroconiferyl ferulate reduces the mRNA levels of c-Myc. (**C**) The expression level of c-Myc in total protein from MDA-MB-231 mammospheres treated with dihydroconiferyl ferulate (50 µM) for 48 h. Dihydroconiferyl ferulate reduces the expression of c-Myc. (**D**) The nuclear protein level of c-Myc was determined in mammospheres derived from MDA-MB-231 cells treated with dihydroconiferyl ferulate (50 µM) for 48 h. Dihydroconiferyl ferulate decreases the level of c-Myc in mammospheres. (**E**) The protein levels of Stat3 and c-Myc in MDA-MB-231 cells treated with a specific Stat3 siRNA. Knockdown of Stat3 decreased the expression of c-Myc. (**F**) qRT-PCR analysis of c-Myc in MDA-MB-231 cells treated with a specific Stat3 siRNA. (**G**) The protein levels of Stat3 and c-Myc in MDA-MB-231 cells treated with a specific c-Myc siRNA. Knockdown of c-Myc does not affect the expression of Stat3. Knockdown of c-Myc inhibits mammosphere formation. Representative data were collected and the data represent the mean ± standard deviation; ** *p* < 0.01 vs. control.

**Figure 8 pharmaceuticals-15-00664-f008:**
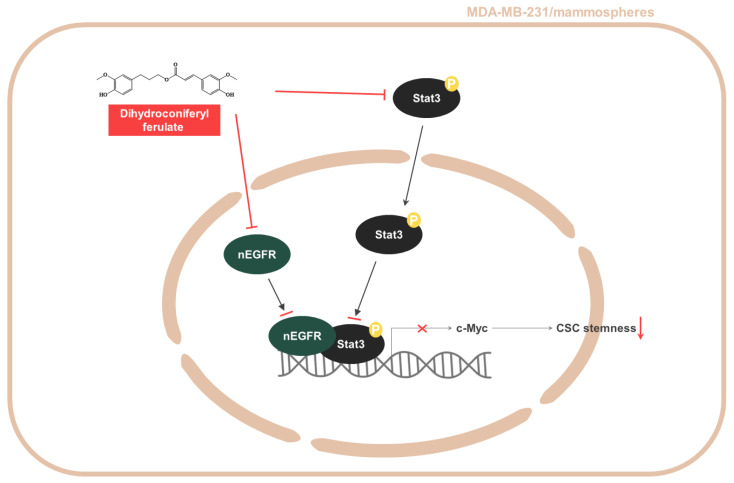
Schematic summary of breast CSC formation via the Stat3/nEGFR/c-Myc signaling pathway induced by dihydroconiferyl ferulate.

## Data Availability

All data is contained within article and Appendix A. In addition, the datasets generated in this study are available upon request to the corresponding author.

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
