# Peer review of "Dihydroconiferyl Ferulate Isolated from Dendropanax morbiferus H.Lév. Suppresses Stemness of Breast Cancer Cells via Nuclear EGFR/c-Myc Signaling"

_pharmaceuticals, 2022, doi:10.3390/ph15060664_

Round 1

Reviewer 1 Report

The authors of this manuscript prove a good knowledge of the issues addressed. We congratulate them for the study and for choosing the research topic. From a medical point of view, this is extremely important in medical research, with the goal of obtaining relevant information in anticancer medication and, moreover, the purification and isolation of natural compounds with anticancer action. The methods are well described, the results and the dilutions clearly highlighting the achievement of the research objective.

We recommend minor revisions to text editing as follows:

- description of all abbreviations at their first appearance in the abstract and in the main text and their subsequent use in the text. For example, on page 1 lines 20 and 25 (CSC — cancer stem cells; and EGFR – epidermal growth factor receptor); also, review aleatory use of abbreviation for dihydroconiferyl ferulate (DHCF);

- Italic throughout the text for the word Araliaceae (written in Latin, defining the family of Dendropanax morbiferus);

- Figure 1/C – defining the axes with the visible measurement units;

- Bold letters for supplementary material captions.

Author Response

Dear Reviewer1

Thank you for your comments concerning our manuscript entitled “Dihydroconiferyl Ferulate Isolated from Dendropanax morbiferus H.Lév. Suppresses Stemness of Breast Cancer Cells via Nuclear EGFR/c-Myc Signaling”.

We greatly appreciate your positive comments. Those comments are all valuable and very helpful for revising and improving our paper, as well as the important guiding significance to our research. We have studied the comments carefully and have made a correction which we hope meets with approval. Revised portions are marked in blue on the manuscript. 

Reviewer 2 Report

Below is my report.

While searching for your plant name, “‪Dendropanax morbiferus” does not appear on the plants of the world database. Instead, I found Dendropanax morbifer H.Lév.

The issue here is that you didn’t write the “author citation” so I’m not sure about the correct name of your plant.

You didn’t mention in the methods how you identified the plant.

A lot of ambiguity about this point. I hope you provide solid info supporting the plant identification.

Line 33-34 a reference is required

The aims of your study are not well described, just broad sentences.

If the name of the plant used in this study is the same as provided, please consider adding more detail to the anticancer and other activities, e.g., the data in this paper “https://doi.org/10.1016/j.foodchem.2013.05.021”

Also, consider describing the novelty of your work in comparison with this study on the same plant using the MCF7 and MDA-MB-231 cells “https://doi.org/10.15204/jkobgy.2015.28.2.026”

About Figure 1A and the extraction procedure 4.3.

I suggest you add a subheading to describe each step individually

The figure and the protocol are poorly described.

As for the protocol, some details are missing or incorrect.

The extraction procedure details (type of extraction, time, temperature..). How you proceeded with the filtration? Did you collect all extract from the 25 flasks and evaporate the solvent? How was the evaporation done? Methanol's boiling point is 64.7 °C. How were you sure that it was completely evaporated? The ethyl acetate fraction was extracted using what? What is the yield obtained? How the EA fraction was concentrated?

In light of those missing info, the protocol should be carefully rewritten along with the descriptive figure.

As for the purification, the protocol is incomplete. Details about the used column ratio of the extract/silica gel used and the other used equipments.

Same goes for the last identification part.

Please subdivide 4.3 into 3 subheads, describe in detail the used method, and support the descriptions with better schemes.

Why IC50 is not calculated

Section 2.2.

Poor description of the results. Instead of giving the obtained results directly, the authors should give more details about the assays and the significance of all figures supporting the results. You cannot refer each time to the figure to close the description of the results.

Figure3A and B * compared to what?

Describing the results of figure 3E and 3F with “‪As shown in Figure 3E and 3F, 50 μM of

‪DHCF inhibited colony formation and migration of MDA-MB-231 and MCF-7 cells.” ‪While the figure has more details to tell is not accepted.  Same goes for the other sections

** and *** compared to what? Please review the statistical description everywhere

Since you poorly described your results when combined. Please split section 2.2. into 3 subheadings and take your time highlighting your findings.

A positive control value is needed to support this sentence “The results indicate that DHCF is a strong suppressor of breast cancer cell migration and colony formation”.

“The structure of DHCF is similar to that of butein and curcumin” a figure to support that will be appreciable

Consider discussing your finding with the paper I mentioned earlier https://doi.org/10.15204/jkobgy.2015.28.2.026” and “https://doi.org/10.1016/j.foodchem.2013.05.021”

Most of your discussion is a paraphrase of the sentences of your results. Not acceptable. You are supposed to discuss your obtained results and highlight their significance, not just re-stating what was obtained in the results.

Despite all the good assays and results obtained in your study. The way the paper is written made it less quality in terms of data processing and overall presentation.

Consider seriously rewrite your paper.

Author Response

Dear Reviewer 2

Thank you for your comments concerning our manuscript entitled “Dihydroconiferyl Ferulate Isolated from Dendropanax morbiferus H.Lév. Suppresses Stemness of Breast Cancer Cells via Nuclear EGFR/c-Myc Signaling”.

We greatly appreciate your good comments. Those comments are all valuable and very helpful for revising and improving our paper, as well as the important guiding significance to our research. We have studied the comments carefully and have made a correction which we hope meets with approval. Revised portions are marked in blue on the manuscript. 

Reviewer 3 Report

The manuscript by Yu-Chan Ko et al. entitled “Dihydroconiferyl ferulate isolated from Dendropanax morbiferus suppresses stemness of breast cancer cells via nuclear EGFR/c-Myc signaling”, aimed to investigate the hinibitory effect of Dihydroconiferyl ferulate on target proteins associated with breast cancer.
The article is really well writtenand the authorsmade a great job even though some imperfections require the authors' attention.
Some corrections should be made in the results and in the Materials and Methods section. See comments below:

  • Please put in the correct order the figures of the Supplementary materials in the paper and in the Supplementary Materials file since the first one that the authors have mentioned in the paper is Figure S5 and into the file there is Table S1 as first one.
  • Please merge paragraphs 2.1 and 2.2 since all the information are closely related
  • Paragraph 2.4: can you please provide a low exposed image of total EGFR protein expression (Figure 5A)?
  • Paragraph 4.6: Please indicate the number of the cells seeded into 96 in the line 350 and cancel the sentence in lines 354-356 " MDA-MA-231...ferulate"

Author Response

Dear Reviewer 3

Thank you for your comments concerning our manuscript entitled “Dihydroconiferyl Ferulate Isolated from Dendropanax morbiferus H.Lév Suppresses Stemness of Breast Cancer Cells via Nuclear EGFR/c-Myc Signaling”.

We greatly appreciate your positive comments. Those comments are all valuable and very helpful for revising and improving our manuscript, as well as the important guiding significance to our research. We have studied the comments carefully and have made a correction which we hope meets with approval. Revised portions are marked in blue on the manuscript. 

Round 2

Reviewer 2 Report

No further comments. The authors amemded the paper correctly following the suggestions.